# Pea Albumin Attenuates Dextran Sulfate Sodium-Induced Colitis by Regulating NF-κB Signaling and the Intestinal Microbiota in Mice

**DOI:** 10.3390/nu14173611

**Published:** 2022-09-01

**Authors:** Shucheng Zhang, Wenhua Jin, Weibo Zhang, Fazheng Ren, Pengjie Wang, Ning Liu

**Affiliations:** 1Key Laboratory of Precision Nutrition and Food Quality, College of Food Science and Nutritional Engineering, China Agricultural University, Beijing 100083, China; 2Beijing Advanced Innovation Center for Food Nutrition and Human Health, Department of Nutrition and Health, China Agricultural University, Beijing 100193, China

**Keywords:** pea albumin, colitis, gut microbiota, inflammation, cytokines

## Abstract

Background: Inflammatory bowel disease remains a global burden with rapidly increasing incidence and prevalence in both industrialized countries and developing countries. In this study, we prepared pea albumin from pea seeds and determined its beneficial effects being anti-inflammatory and on gut microbiota modulation in dextran sulfate sodium (DSS)-challenged mice. Method: Six-week-old C57BL/6N male mice received an equivalent volume (200 μL) of sterile phosphate balanced solution, 0.375, 0.75, or 1.50 g/kg body weight (BW) of pea albumin that was subjected to 2.0% DSS for 7 days to induce colitis. On day 17 of the experiment, all mice were sacrificed after blood sample collection, and colon tissue and colon contents were collected. BW change curve, colon length, myeloperoxidase (MPO) activity, mucus staining, immunofluorescence staining of T cells and macrophages, cytokines, pro-inflammatory genes expression, nuclear factor-κB (NF-κB) and signal transducer, and activator of transcription 3 (STAT3) signaling pathways as well as 16S DNA sequence were measured. Results: Our results show that pea albumin alleviates DSS-induced BW loss, colon length shortening, enhanced MPO activity, cytokines secretion, mucus deficiency, and inflammatory cell infiltration, as well as enhanced pro-inflammatory genes expression. In addition, the overactivation of NF-κB and STAT3 following DSS exposure is attenuated by pea albumin administration. Of particular interest, pea albumin oral administration restored gut microbiota dysbiosis as evidenced by enhanced α-diversity, restored β-diversity, and promoted relative abundance of *Lactobacillus* and *Lachnospiraceae_NK4A136_group*. Conclusion: Taken together, the data provided herein demonstrated that pea albumin plays a protective role in DSS-induced colitis by reducing inflammatory cell infiltration, pro-inflammatory genes expression and pro-inflammatory cytokines release, inactivation of NF-κB signal, and gut microbiota modulation.

## 1. Introduction

Epidemiological and observational studies indicate that inflammatory bowel disease (IBD) has become a global burden with rapidly increasing incidence and prevalence in both industrialized countries and developing countries [1,2]. IBD, including ulcerative colitis (UC) and Crohn’s disease (CD), is a noninfectious, chronic, and relapsing inflammatory disorder of the gastrointestinal tract in which the interactions among immune responses, barrier function, nutrition, and gut microbiome are involved [3,4]. Although extensive attempts have been made, the precise etiology and pathogenesis of IBD have not yet been defined [5]. In general, it is believed that both genetic and environmental factors (nutrition and intestinal microbiota) contribute to the initiation or progression of IBD [6]. Multiple symptoms, including diarrhea, abdominal pain, cramping, rectal bleeding, weight loss, spontaneous remission, bloody stool, and relapsing inflammation, are observed over time and impair the patients’ quality of life [7,8]. Furthermore, excessive activation of nuclear factor-κB (NF-κB) and signal transducer and activator of transcription 3 (STAT3), critical transcript factors involved in the synthesis and release of cytokines and innate immune response, were observed in the inflamed intestinal mucosa [9]. In addition, growing evidence indicates that gut microbiota dysbiosis is observed in IBD and contributes to initiation or progression of IBD [8,10]. Thus, strategies that aim at inhibiting NF-κB and STAT3 signaling pathways or restoring gut microbial community might be potentially useful therapeutic applications to alleviate IBD.

Epidemiological data have shown that a diet characterized by high protein and unsaturated fats from animals, high sugar, and low fiber derived from fruits and vegetables can trigger a pro-inflammatory response in susceptible individuals [11,12], indicating a more significant influence of nutrition on IBD development. Furthermore, protein from animal origin has been associated with a general harmful microbiota profile, whereas protein from plant origin is positively associated with microbial diversity [13]. Dietary components may exert direct effects on the expression of transcription factors involved in the inflammatory response and balance of the intestinal microbiota composition [14,15]. Furthermore, dietary components may also indirectly affect immune response by affecting the composition and function of gut microbiota [16]. In consequence, growing evidence indicates that nutritional interventions play an important role in the treatment and management of IBD or other gastrointestinal diseases through modulating intestinal homeostasis [11,12]. Interestingly, accumulating studies show that Bowman–Birk inhibitors (BBIs) in legumes exert potential anti-inflammatory effects on DSS-induced colitis [17]. Furthermore, a randomized double-blind placebo-controlled trial demonstrated that patients with active UC who were treated with soybean BBI showed improvements in multiple symptoms, including disease activity index (DAI) score, stool frequency, and rectal bleeding [18]. Pea albumin as one of four major groups of pea proteins (globulin, albumin, prolamin, and glutelin) accounts for approximately 18–25% of total pea protein and is rich in bioactive compounds including BBI, pea albumin 1, and pea albumin 2 [19,20,21,22]. In this regard, we hypothesized that pea albumin oral administration could have a positive effect on colitis alleviation. In the present study, mice supplemented with different doses of pea albumin by oral gavage were subjected to a 2% DSS solution to induce acute colitis. The underlying mechanism of pea albumin in protecting against colitis was investigated.

## 2. Materials and Methods

### 2.1. Reagents

The DSS salt (MW 36–50 kDa) was purchased from MP Biomedicals (Santa Ana, CA, USA). Primary antibodies against NF-κB (#8242), phosphor (p)-NF-κB p65 (Ser536, #3033), STAT3 (9139), and p-STAT3 (Tyr705, 9145), were obtained from Cell Signaling Technology (Danvers, MA, USA). Antibodies against GAPDH were purchased from Santa Cruz Biotechnology (San Diego, CA, USA). Peroxidase-conjugated secondary antibodies against rabbit and mouse and an enhanced chemiluminescence (ECL) kit were obtained from Huaxingbio Biotechnology Co. (Beijing, China). The bicinchoninic acid protein (BCA) assay kit was purchased from Huaxingbio Biotechnology Co. (Beijing, China). Polyvinylidene fluoride (PVDF) membranes were purchased from Millipore (Billerica, MA, USA). Kits for the detection of myeloperoxidase (MPO) were obtained from Nanjing Jiancheng Bioengineering Institute (Nanjing, China). Serum IL (interleukin)-6 (PI326) and IL-10 (PI522) levels were measured using commercially available kits from Beyotime Biotechnology (Haimen, China) according to the manufacturer’s instructions. IL-17 (KET7012) and IL-22 (KET7013) in colonic tissue were determined using ELISA kits purchased from (EliKine™, Abbkine, Wuhan, China), and ELISA was performed according to the manufacturer’s instructions. An Alcian blue (AB) staining kit was purchased from Solarbio Life Sciences (Beijing, China). Unless stated, all other chemicals were obtained from Sigma-Aldrich (St. Louis, MO, USA).

### 2.2. Experimental Animals

All the animal experimental procedures were approved by the Institutional Animal Care and Use Committee of China Agricultural University (AW12402202-5-5). Five-week-old C57BL/6N male mice were purchased from the Beijing Vital River Laboratory Animal Technology Co., Ltd. (Beijing, China) and raised in a controlled environment (24 ± 2 °C, 50 ± 5% humidity, 12 h dark–light cycle) with free access to food and drinking water. After a 1-week adaptation period, mice were randomly assigned to one of the following six groups (*n* = 9 each): Control group; Control_P group (0.75 g/kg body weight pea albumin by oral gavage); DSS group (2.0% *w*/*v* DSS in drinking water); DSS_PL group (2.0% *w*/*v* DSS in drinking water + 0.375 g/kg body weight pea albumin by oral gavage); DSS_PM group (2.0% *w*/*v* DSS in drinking water + 0.75 g/kg body weight pea albumin by oral gavage); and DSS_PH group (2.0% *w*/*v* DSS in drinking water + 1.5 g/kg body weight pea albumin by oral gavage). A detailed program of drug administration throughout the experiment period is presented in Appendix A. On day 17 of the experiment, all mice were sacrificed after blood sample collection via retro-orbital puncture, and colon tissue and colon contents were collected and then stored at –80 °C for later analysis. Body weight (BW) and food intake were recorded daily throughout the experiment.

### 2.3. Preparation of Pea Albumin

Pea seeds were provided by Yantai Shuangta Food Co., Ltd. (Yantai, China). The pea albumin was prepared according to the previous description [23] with slight modifications. Briefly, ground pea seeds were defatted with hexane (1:3 *w*/*v*) and then washed with 85% alcohol (1:10 *w*/*v*). Next, the pea meal was dissolved in distilled water (1:10 *w*/*v*) and stirred for 60 min at pH 9.0. The solution was centrifuged at 10,000 rpm for 30 min and then the collected supernatant was stirred for 30 min at pH 4.5 with the addition of HCl followed by centrifugation at 10,000 rpm for 30 min. The cold supernatant was centrifuged at 10,000 rpm for 30 min after storing at 4 °C for 24 h. The collected solution was dialyzed against distilled water for 48 h at 4 °C. After that, the dialyzed extract was subjected to 80% ammonium sulfate for salting out. The coacervate was collected after centrifugation at 10,000 rpm for 30 min and fully dissolved in water. After dialyzing against distilled water, the dialyzed extract was lyophilized and stored at −20 °C for later analysis.

### 2.4. Sodium Dodecyl Sulfate–Polyacrylamide Gel Electrophoresis

Pea albumin was characterized using SDS–PAGE to monitor the electrophoretic pattern following a previous method [24]. Briefly, the lyophilized powder of pea albumin was dissolved in distilled water to keep the concentration at 0.2% (*w*/*v*). The sample (8 μL) mixed with 2 μL dithiothreitol was boiled for 5 min and then loaded on to the gel. The electrophoresis was run for 40 min at a constant voltage of 30 V in 4% stacking gel and then at a constant voltage of 100 V in 12% separating gel for 60 min. SDS–PAGE analysis was performed using Quantity One version 4.6.3 software (Bio-Rad, Madrid, Spain) after staining with a Coomassie Blue staining kit.

### 2.5. Composition Analysis of Pea Albumin

Composition analysis was carried out using diverse analytical methods. The contents of protein, ash, moisture, fat (total fat, saturated fat, and trans fat), cholesterol, dietary fiber (total dietary fiber, soluble fiber, and insoluble fiber), and vitamin D were determined according to AOAC 979.09, AOAC 923.03, AOAC 935.29, AOAC 996.06, AOAC 994.10, AOAC 991.43, and AOAC 982.29-1983, respectively. The contents of lead (Pb), arsenic (As), mercury (Hg), cadmium (Cd), and chromium (Cr) were analyzed based on GB 5009.12-2017, GB 5009.11-2014, GB 5009.17-2014, GB 5009.15-2014, and GB 5009.123-2014, respectively. The amino acids profile of pea albumin was measured using high-performance liquid chromatography methods as previously described. Total carbohydrate content and energy were calculated based on FDA 21 CFR 101.9.

### 2.6. Disease Activity Index Assessment and Colon Length

The DAI was determined by calculating the BW loss score, stool consistency score, and fecal blood content score according to a previous description [6]. At the end of the experiment, the mice were sacrificed and then the colon was immediately dissected. The length of the colon was measured with a ruler.

### 2.7. Colonic Myeloperoxidase Activity

The activities of MPO in colon tissue were determined using commercial kits (Nanjing Jiancheng Bioengineering Institute). Data were expressed as U/g tissue weight.

### 2.8. Histopathological Analyses

Fixed colonic tissues were sectioned and stained with H&E (ZSGB, Beijing, China). At least six slides were randomly visualized using a light microscope equipped with a computer-assisted morphometric system by a blinded observer, and representative images were photographed. Histopathological scores were determined by intestinal surface, crypt destruction, and inflammatory cell infiltration [25] (Appendix A).

For mucus analysis, colonic paraffin sections were stained with AB, according to the manufacturer’s instructions. Mucin was observed with a light microscope. Stained images were observed by different blinded investigators.

### 2.9. Immunofluorescence Staining

Immunofluorescence staining was performed to determine inflammatory cell infiltration as previously described [10]. The colonic sections were incubated with primary antibodies CD4, CD8, CD11b, or F4/80 overnight at 4 °C, and then followed by incubation with fluorescein-labeled secondary antibodies for 1 h at room temperature. Nuclei were stained with Hoechst 33342 for 3 min. Isotype controls were performed as negative controls. Sections were visualized and analyzed by a blinded observer using TCS SPE fluorescent microscopy (Leica, Wetzlar,, Germany).

### 2.10. Detection of IL-6, IL-10, IL-17, and IL-22

Serum IL-6 and IL-10 were detected using enzyme-linked immunosorbent assay (ELISA) kits according to the manufacturer’s instructions. IL-17 and IL-22 contents in the colonic tissue were determined by using ELISA kit according to the manufacturer’s instructions.

### 2.11. Quantitative Real-Time PCR

Total RNA was isolated from colonic tissue using TRIzol reagent following the manufacturer’s instructions, and then reverse transcribed by a FastQuant RT kit with gDNase (TIANGEN Biotech, Beijing, China). Quantitative real-time PCR was performed using SYBR green mix and a real-time PCR detection system (ABI 7500 system, Applied Biosystems, Foster, CA, USA). The primer sequences used in the present study are listed in Appendix A. Relative gene expression was determined using 2^−ΔΔCt^ method. GAPDH was used as an internal control.

### 2.12. Western Blot Analysis

Total proteins from colon tissue were isolated using RIPA lysis buffer (10 mM Tris-HCl, pH 7.4; 150 mM NaCl; 10 mM EDTA; 1% NP-40; 0.1% SDS; 1.0 mM PMSF; 1.0 mM Na_3_VO_4_; 1.0 mM NaF) supplemented with protease and phosphatase inhibitors. The concentration of protein was quantified using a BCA protein assay kit (Huaxingbio Bio-technology Co., Beijing, China). Identical amounts of proteins (50 μg) were separated on SDS–PAGE gels, transferred to PVDF membranes (Millipore), and blocked with 5% skim milk for 1 h at room temperature. After incubation with a primary antibody overnight at 4 °C, the membranes were washed with buffer and incubated with a horseradish peroxidase (HRP)-conjugated secondary antibody at room temperature for 1 h. The protein bands were incubated with an ECL kit and visualized using an ImageQuant LAS 4000 mini system (GE Healthcare, Piscataway, NJ, USA). Band density was measured using the ImageJ software (NIH, Bethesda, MD, USA). GAPDH was used as a loading control.

### 2.13. Gut Microbial Analysis

Gut microbiota DNA was extracted from colon content collected in a sterile environment using FastDNA^®^ SPIN for soil kit (MP Biomedicals, Solon, OH, USA) according to the manufacturer’s recommendations. The 16S rDNA V3-V4 region of the bacterial gene was amplified with universal primers 338F (ACTCCTACGGGAGGCAGCAG) and 806R (GGACTACHVGGGTWTCTAAT). The amplicons’ products were pooled at equal concentrations and paired-end sequenced on an Illumina MiSeq PE300 platform (Illumina, San Diego, CA, USA) for high-throughput sequencing Magigene Technology Co., Ltd. (Guangzhou, China). Chimeras were filtered using USEARCH, and the remaining sequences were grouped to generate operational taxonomic units with 97% similarity cutoff using UPARSE algorithm [26]. The 16S rRNA gene sequence data were processed using linear discriminant analysis effect size (LEfSe). LEfSe differences among biological groups were tested for significance using a nonparametric factorial Kruskal–Wallis sum-rank test followed by Wilcoxon rank-sum test. A principal coordinate analysis plot was created using the R software based on Bray–Curtis dissimilarity.

### 2.14. Statistical Analysis

All results are presented as mean ± SEM. Multiple comparisons among groups were evaluated using one-way analysis of variance followed by Duncan’s multiple comparison method with SAS software, version 9.1 (SAS Institute Inc., Cary, NC, USA). *p* < 0.05 was considered statistically significant.

## 3. Results

### 3.1. Composition of Pea Albumin

To determine the protein pattern and molecular weight of pea albumin, SDS–PAGE analysis was employed. As illustrated in Figure 1, four major bands were detected clearly in pea albumin, which correspond to ~10, 15, 24, and 100 kDa.

As listed in Table 1, pea albumin was composed of 79.7% protein, 6.10% ash, 4.95% moisture, 0.25% total fat, 9.0% total carbohydrate, and 5.59% total dietary fiber. The total dietary fiber composition of pea albumin included 4.86% soluble fiber and 0.73% insoluble fiber. The analytical results of heavy metals in pea albumin (Table 1) showed that the concentrations of Pb, As, Cd, and Cr are 0.17, 0.019, 0.04, and 2.35 mg/kg, respectively. In addition, trans fat, cholesterol, and Hg in pea albumin were not detected (Table 1). Furthermore, the amino acid (AA) composition of pea albumin was analyzed and is shown in Table 1. Lysine, threonine, arginine, and valine were the most abundant among the essential AAs, whereas glutamate and aspartate were the most abundant among the nonessential AAs.

### 3.2. Pea Albumin Alleviated DSS-Induced Colitis

Mice challenged with 2.0% DSS in water exhibited obvious symptoms (*p* < 0.05) of colon injury as indicated by reduced BW (Figure 2A), increased DAI score (Figure 2B), shortened colon length (Figure 2C,D), and appearance of diarrhea and bloody stool (Figure 2C) compared with the control mice. In addition, mice challenged with DSS had elevated (*p* < 0.05) MPO activity (Figure 2E) in the colon tissue, an indicator of inflammation. These adverse changes were significantly ameliorated (*p* < 0.05) by pea albumin at a dose of 0.75 g/kg BW, but not at 0.375 or 1.5 g/kg BW. These data suggested that 0.75 g/kg BW pea albumin oral administration effectively alleviated DSS-induced colon injury in mice.

### 3.3. Pea Albumin Attenuated Inflammatory Cell Infiltration in Colon Tissue

Histological changes and mucus abundance were analyzed using H&E and AB staining, respectively. As shown in Figure 3A–C, DSS treatment resulted in impaired crypts, colonic epithelium loss, enhanced histological scores, inflammatory cell infiltration, and reduced mucus abundance. Furthermore, immunofluorescence staining demonstrated an increased number of T-lymphocytes (CD4^+^, CD8^+^, Appendix A) and macrophages (CD11b^+^, F4/80^+^, Appendix A) in the colon of DSS-challenged mice. Interestingly, mice treated with pea albumin showed obvious attenuation of colonic damage and inflammatory cell infiltration in DSS-challenged mice, indicating a protective role for pea albumin on the colonic mucus barrier.

### 3.4. Pea Albumin Ameliorated DSS-Induced Secretion of Inflammatory Cytokines and Overactivation of STAT3 and NF-κB

To examine the regulatory effects of pea albumin on DSS-induced inflammatory response, IL-6 and IL-10 concentrations in serum, IL-17 and IL-22 concentrations in colon tissue, pro-inflammatory genes expression, and inflammation-associated signaling pathways were determined. As shown in Figure 4A,B, elevated protein level of IL-6 in serum and elevated protein level of IL-17 and IL-22 in colon tissue were found in the DSS group when compared with the control group (*p* < 0.05), but no difference was obtained in IL-10 concentrations in serum (*p* > 0.05). Pea albumin administration markedly (*p* < 0.05) reduced the IL-6 in serum and IL-17 and IL-22 concentrations in colon tissue. In addition, DSS treatment led to enhanced (*p* < 0.05) mRNA expression of proinflammatory cytokine genes expression compared with the control, including TNF-α, IFN-γ, IL-1β, IL-6, IL-17A, and IL-22. However, these effects were greatly abrogated by pea albumin administration (Figure 4C). Furthermore, Western blot analysis showed that DSS treatment markedly enhanced (*p* < 0.05) protein abundances of p-STAT3 and p-NF-κB in the colon (Figure 4D–F), and this effect was abolished by coadministration with pea albumin (*p* < 0.05). These results indicated that pea albumin exerts anti-inflammatory effects via suppressing the inactivation of the NF-κB and STAT3 signaling pathways.

### 3.5. Pea Albumin Restored Gut Microbiota Community

To confirm whether the alleviation of DSS-induced colitis using pea albumin was attributed to a change in gut microbiota composition, a 16S rDNA sequencing analysis of gut microbiota was performed. As shown in Figure 5A,B, the Chao1 index (*p* < 0.05) and Shannon index were diminished by DSS treatment compared with control mice, which were restored (*p* < 0.05) by pea albumin administration (Figure 5A,B). β-Diversity analyses using principal component analysis (PCA) displayed a marked separation of gut microbial structure between control and DSS groups (Figure 5E). According to PCA and ANOSIM (R^2^ = 0.547, *p* < 0.001), DSS-treated mice with pea albumin administration shaped partially overlapping clusters in the ordination plot close to the control group or control_P group (Figure 5E), suggesting that DSS treatment changed the gut microbiota structure, and pea albumin cotreatment exerted modulatory effects. As illustrated in Figure 5C, Firmicutes, Bacteroidetes, Verrucomicrobia, Tenericutes, and Proteobacteria represent the predominant gut microbiota at the phylum level. DSS treatment led to a higher abundance of Verrucomicrobia when compared with the control group. At the genus level, DSS treatment reduced the relative abundance of *Lactobacillus* (Figure 5D) and enhanced the relative abundance of *Akkermansia* (Figure 5D) when compared with control mice. Of note, these values tended to restore to control mice through the intervention of pea albumin (Figure 5D). Next, LEfSe analysis was performed to determine differentially abundant fecal bacterial taxa in DSS-treated mice in response to pea albumin administration. As shown in Figure 5F,G, we found that eight feature bacterial genera, particularly including *Lactobacillus,* were enriched by pea albumin administration (Appendix A), whereas the other seven feature bacterial genera were enriched in the DSS group. Above all, these data suggest that pea albumin restored the gut microbiota in DSS-treated mice.

### 3.6. Spearman Correlation between Intestinal Microbiota and Parameters

Furthermore, Spearman correlation analysis was performed to understand the correlation coefficients between differentially enriched microbes and DSS-associated traits (BW, MPO, IL-6, IL-10, colon length, etc.). The results in Figure 6 indicate that *Akkermansia* displayed a strong negative correlation with BW (*p* < 0.05) and colon length (*p* < 0.05). *Lactobacillus* showed a strong negative correlation with IL-6 and MPO and a strong positive correlation with BW (*p* < 0.05) and colon length (*p* < 0.05). These results show that pea albumin administration enhanced *Lactobacillus* abundance, which plays a critical role in alleviating DSS-induced colitis.

## 4. Discussion

In the present study, a protective role of pea albumin against DSS-induced colitis, a well-known model for IBD, was investigated. Our results demonstrated that beneficial effects of pea albumin were associated with reduced inflammatory cell infiltration (T-lymphocytes and macrophages); reduced secretion of IL-6, IL-17, and IL-22; reduced pro-inflammatory genes expression; abrogation of excessive activation of NF-κB and STAT3 signaling pathways; as well as selectively enhanced the abundance of *Lactobacillus* in the colon.

SDS–PAGE was employed to determine the protein pattern and molecular weight of pea albumin (Figure 1). Four major bands were detected clearly in pea albumin and can be identified as lipoxygenase, pea albumin 2, trypsin inhibitors, and pea albumin 1 based on their molecular weights (~10, 15, 24, and 100 kDa, respectively) as referenced with previous studies of pea albumin protein composition [23,27]. Chemical analysis of the composition of pea albumin is shown in Table 1. The total protein content of pea albumin is 79.70%, which agrees with a previous report [20]. In pea albumin, lysine (8.07%), threonine (4.73%), arginine (4.58%), and valine (3.43%) were the most abundant among the essential AAs, whereas glutamate (12.18%) and aspartate (9.42%) were the most abundant among the nonessential AA. These values on AA composition obtained here are similar to previous reports [19,20]. Pb, As, Cd, Cr, and Hg are known as the most problematic heavy metals and can exert toxic effects at very low concentrations [28]. The analytical results of heavy metals in pea albumin (Table 1) showed that the concentrations of Pb, As, Cd, and Cr are 0.17, 0.019, 0.04, and 2.35 mg/kg, respectively. These values indicated that pea albumin is safe and nutritious.

Epidemiologic evidence and intervention trials indicate that consumption of some plant proteins may exert antioxidant, antihypertensive, and anti-inflammatory effects [21,22,29,30,31,32]. Pea albumin, a relatively new type of plant protein with a well-balanced AA profile, accounts for approximately 18–25% of total pea protein and is rich in bioactive compounds that may synergistically act to exert anti-inflammatory and gut microbiota modulation [19,20,21,22]. To investigate the underlying mechanism of pea albumin oral administration in alleviating DSS-induced colitis, DSS-challenged mice were pretreated with different doses of pea albumin (0.375, 0.75, or 1.5 g/kg BW) by oral gavage according to our pilot study and a previous study [17]. Pea albumin (0.75 g/kg BW) supplementation markedly reversed BW loss, shortened colon length, increased DAI scores, and enhanced MPO activity induced by DSS treatment (Figure 2). The protective effects of pea albumin against DSS-induced colitis may be attributed to the anti-inflammatory effects of BBIs within the large intestine [17]. Consistently, soybean BBI exerted beneficial effects when applied to DSS-induced colitis [33]. Furthermore, preclinical studies and experimental animal studies suggested that an extract enriched in soybean BBI reduced inflammatory processes in extent and severity without any adverse side effects [18,34]. Few studies have determined the anti-inflammatory effect of pea albumin 1 and pea albumin 2 on colitis; thus, further studies are needed to understand the potential contribution of the different pea albumins.

Colonic mucus layer deficiency and inflammatory cell infiltration (T-lymphocytes, macrophages, etc.) have been characterized in DSS-induced mice [10], which was also observed in our present study. The mucus layer in the intestinal epithelium is an important barrier to prevent microbes from invading the colonic mucosa [35]. In the present study, we found that DSS treatment reduced the mucus deficiency presented by AB staining, which is consistent with previous reports [36]. Oral administration of pea albumin attenuated the mucus deficiency induced by DSS treatment, indicating a protective role of pea albumin on mucosal architecture and abundance. A previous study also revealed that pea seed albumin extract increased the mRNA expression of MUC2 and MUC3, as well as a tight junction, contributing to the intestinal barrier integrity [17,37]. Intestinal inflammation response was commonly associated with an impaired intestinal barrier leading to the entry of antigens into the colonic mucosa [38]. Of note, histopathological analyses and immunofluorescence staining demonstrated that pea albumin administration to DSS-challenged mice reduced the abundance of T-lymphocytes and macrophages in colon tissue. The improvement in mucus abundance and barrier may contribute to the inhibition of inflammatory cells infiltration and modulation of the immune response.

IL-6, IL-17, and IL-22 pro-inflammatory cytokines were elevated in DSS-challenged mice and was suppressed by pea albumin administration. Moreover, pea albumin numerically enhanced IL-10 concentrations with no significance in serum when compared with DSS-challenged mice. In line with previous reports [8,10,39], evaluated mRNA expression of TNF-α, IFN-γ, IL-1β, IL-6, IL-17A, and IL-22 were observed in DSS-challenged mice. These effects were reduced by pea albumin administration suggesting the anti-inflammatory effects of pea albumin. IL-6 and IL-22 have been reported to activate STAT3 signaling pathway [40,41], whose activation are associated with the innate immune responses. Beneficial effects were attributed to the serine protease′s inhibition of pea albumin [42]. Both NF-κB and STAT3 are pivotal transcription factors implicated in the inflammatory response and cytokines secretion and release [6,43]. Overactivation of NF-κB and STAT3 was observed in colitis mice, which agrees with previous studies [41,44]. Conversely, inhibition of NF-κB and/or STAT3 signaling pathways has been reported to reduce pro-inflammatory cytokines and inflammation response in colitis [45,46]. Of particular interest, our results found that pea albumin significantly alleviated the upregulation of phosphorylation of NF-κB and STAT3, indicating that pea albumin contains bioactive compounds with the ability to block the NF-κB and STAT3 signaling pathways in DSS-induced colitis. These beneficial effects of pea albumin may account for vegetable intake being associated with a lower risk of IBD than red meat intake [47].

Gut microbiota dysbiosis and its contribution to the initiation or progression of IBD have promoted the active search for novel therapeutic interventions by targeting the gut microbiota [48,49]. In the present study, DSS treatment led to gut microbiota dysbiosis revealed by reduced α-diversity (Chao1 index and Shannon_10 index) and changed β-diversity of PCA (microbiota community structure), consistent with previous studies [50,51]. Pea albumin administration was found to increase the α-diversity and restore the β-diversity of gut microbiota, suggesting the potential role of pea albumin in restoring the intestinal microbial community. Notably, pea albumin reversed the reduction in the relative abundance of *Lactobacillus* and *Lachnospiraceae_NK4A136_group* (Appendix A), which have been reported to alleviate inflammatory response and promote the repair of the intestinal mucosa [10,52,53]. Conversely, pea albumin reversed the increase in the relative abundance of *Akkermansia*, which is consistent with a previous study [54]. In general, *Akkermansia* is a commensal bacterium of the gut, contributing to the mucosal and systemic immunity, whereas they will escape into the peritoneum when disruptions occur in the colon mucus [54], as demonstrated by colonic mucus layer deficiency in DSS-challenged mice in our study. Thus, these data indicate that pea albumin protects mice from colitis by restoring gut microbiota and selectively enhanced the relative abundance of beneficial intestinal bacteria (*Lactobacillus* and *Lachnospiraceae_NK4A136_group*, etc.).

## 5. Conclusions

In conclusion, we found that pea albumin administration by oral gavage mitigated DSS-induced colitis. This beneficial effect of pea albumin was associated with inhibition of pro-inflammatory cytokines’ (IL-6, IL-17, and IL-22) release, abrogation of overactivation of NF-κB and STAT3 as well as restored microbiota structure and composition. Further studies are necessary to understand the contribution of different components of pea albumin in colitis alleviation.

## Figures and Tables

**Figure 1 nutrients-14-03611-f001:**
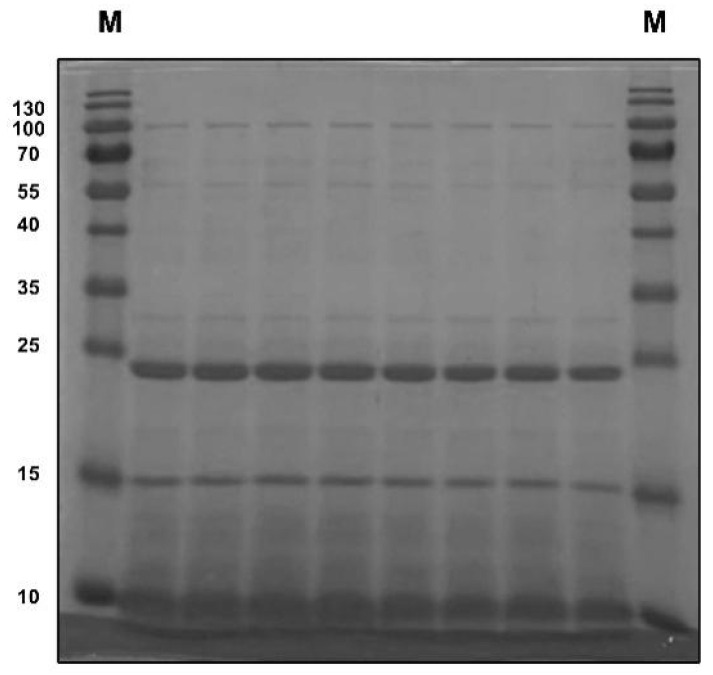
Sodium dodecyl sulfate–polyacrylamide gel electrophoresis for pea albumin.

**Figure 2 nutrients-14-03611-f002:**
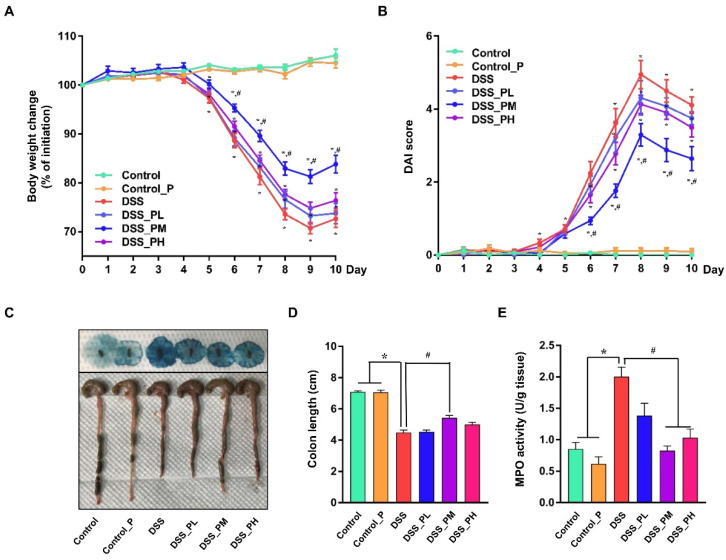
Pea albumin attenuated the severity of DSS-induced colitis. (**A**) Body weight change from day 0 to day 10 and the body weight loss relative to the baseline (day 0). (**B**) DAI score. The sum of the score recorded for weight loss (%), stool consistency, and fecal blood content was presented. (**C**) Representative images of bloody stool (up) and the colon length. (**D**) Statistical analysis of colon length. (**E**) MPO activity (U/g tissue). Data are shown as mean ± SEM. *n* = 9 per group. * *p* < 0.05 versus the control group. ^#^ *p* < 0.05 versus the DSS group. DAI, disease activity index; DSS, dextran sulfate sodium; MPO, myeloperoxidase; Control_P, 0.75 g/kg body weight pea albumin by oral gavage; DSS_PL, 2.0% *w*/*v* DSS in drinking water + 0.375 g/kg body weight pea albumin by oral gavage; DSS_PM, 2.0% *w*/*v* DSS in drinking water + 0.75 g/kg body weight pea albumin by oral gavage; DSS_PH group, 2.0% *w*/*v* DSS in drinking water + 1.5 g/kg body weight pea albumin by oral gavage.

**Figure 3 nutrients-14-03611-f003:**
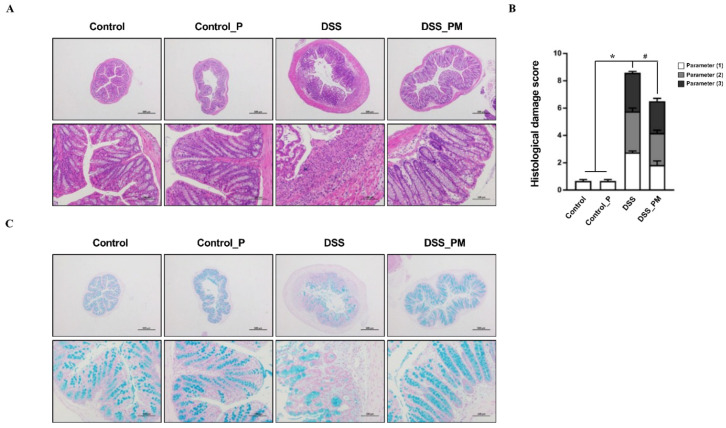
Pea albumin alleviated DSS-induced histological damage and mucus deficiency in colitis. (**A**) Representative H&E staining images of colonic tissue sections. Scale bar means 500 μm (up) and 100 μm (down). (**B**) Histopathological scores. Parameters (1): Loss of epithelial surface; Parameters (2): Destruction of crypt; Parameters (3): Infiltration of inflammatory cells. (**C**) Representative images of AB-stained inner mucus layer of colonic sections. Scale bars indicate 500 μm (up) and 100 μm (down). Data are shown as mean ± SEM. *n* = 9 per group. * *p* < 0.05 versus the control group. ^#^ *p* < 0.05 versus the DSS group. AB, Alcian blue; DSS, dextran sulfate sodium; Control_P, 0.75 g/kg body weight pea albumin by oral gavage; DSS_PM, 2.0% *w*/*v* DSS in drinking water + 0.75 g/kg body weight pea albumin by oral gavage.

**Figure 4 nutrients-14-03611-f004:**
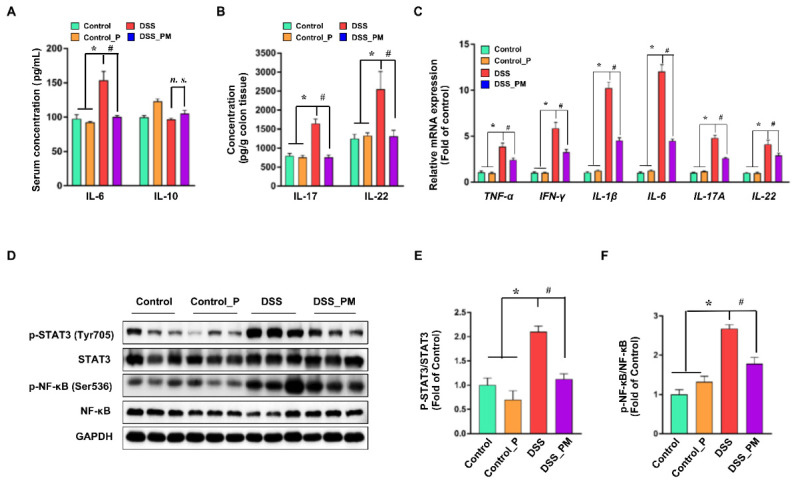
Pea albumin mitigated the induction of inflammatory response. (**A**) The serum protein level of IL-6 and IL-10. (**B**) The protein level of IL-17 and IL-22 in colon tissue. (**C**) Relative mRNA expression of TNF-α, IFN-γ, IL-1β, IL-6, IL-17A, and IL-22 in colon tissue. (**D**) Representative bands of p-STAT3 (Tyr705), STAT3, p-NF-κB (Ser536), NF-κB, and GAPDH in colonic tissue. (**E**,**F**) Statistical analysis of protein abundances in the colon. Data are shown as mean ± SEM. *n* = 9 per group. * *p* < 0.05 versus the control group. ^#^ *p* < 0.05 versus the DSS group. DSS, dextran sulfate sodium; Control_P, 0.75 g/kg body weight pea albumin by oral gavage; DSS_PM, 2.0% *w*/*v* DSS in drinking water + 0.75 g/kg body weight pea albumin by oral gavage; NF-κB, nuclear factor-κB; STAT3, signal transducer and activator of transcription 3.

**Figure 5 nutrients-14-03611-f005:**
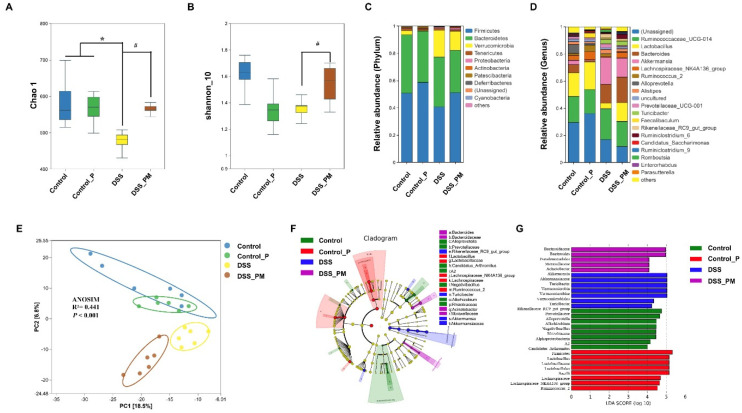
Pea albumin alleviated DSS-induced gut microbiota dysbiosis and reshaped the gut microbiota community. (**A**) Chao1 index. (**B**) Shannon_10 index. (**C**) Relative abundance of gut microbiota at the phylum level. (**D**) Relative abundance of gut microbiota at the genus level. (**E**) β-Diversity of bacteria community based on PCA and ANOSIM analysis. (**F**,**G**) Analysis of differences in the microbial taxa with a linear discriminant analysis score of 4.0 or greater from phylum to genus levels in gut microbiota communities under different treatments (LEfSe difference in dominant microorganisms under different treatments). Data are mean ± SEM, *n* = 6/group. * *p* < 0.05 versus the control group. ^#^ *p* < 0.05 versus the DSS group. DSS, dextran sulfate sodium; Control_P, 0.75 g/kg body weight pea albumin by oral gavage; DSS_PM, 2.0% *w*/*v* DSS in drinking water + 0.75 g/kg body weight pea albumin by oral gavage.

**Figure 6 nutrients-14-03611-f006:**
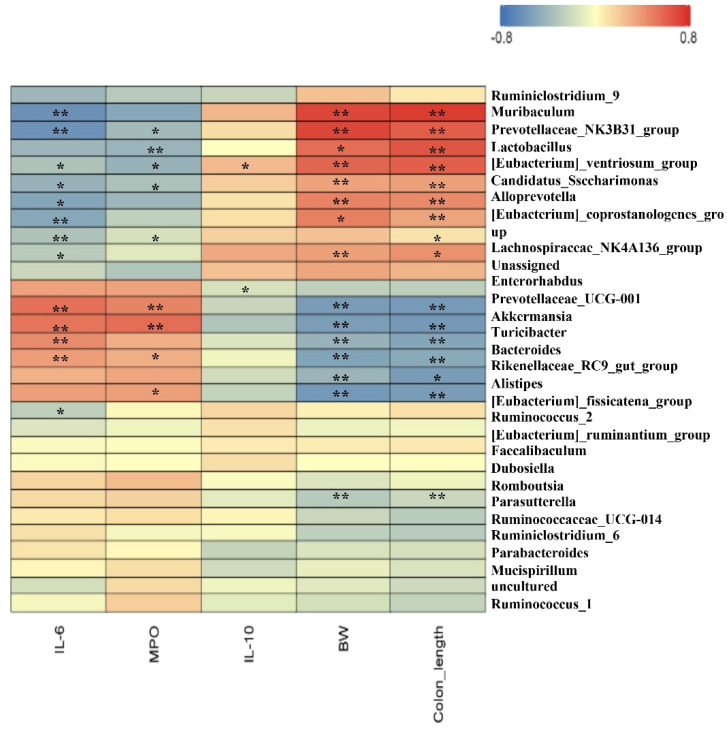
Spearman correlation between gut top 30 genera and growth development or serum biochemical parameters. The red color denotes a positive correlation, whereas the blue color denotes a negative correlation. The intensity of the color is proportional to the strength of the Spearman correlation. Data are mean ± SEM, *n* = 6/group. *, ** indicate a significant different. DSS, dextran sulfate sodium; MPO, myeloperoxidase; BW, body weight.

**Table 1 nutrients-14-03611-t001:** Composition and amino acid profile of pea albumin.

Parameters	Pea Albumin	Essential Amino Acids (%)	
Dry matter (%)	95.05	Lysine	8.07
Protein (%)	79.70	Threonine	4.73
Ash (%)	6.10	Arginine	4.58
Moisture (%)	4.95	Valine	3.43
Fat (%)	0.25	Leucine	2.86
Saturated fat (%)	0.13	Phenylalanine	2.81
Trans fat (%)	ND	Isoleucine	2.44
Cholesterol (%)	ND	Tryptophan	0.84
Carbohydrate (%)	9.00	Methionine	0.78
Dietary fiber (%)	5.59	Nonessential amino acids (%)	
soluble fiber	4.86	Glutamate	12.18
insoluble fiber	0.73	Aspartate	9.42
Energy (kcal/g)	3.57	Alanine	5.85
Vitamin D (μg/kg)	ND	Glycine	5.14
Lead (mg/kg)	0.17	Serine	3.75
Arsenic (mg/kg)	0.019	Proline	3.53
Mercury (mg/kg)	ND	Tyrosine	3.03
Cadmium (mg/kg)	0.04	Histidine	2.45
Chromium (mg/kg)	2.35	Cystine	2.20

Note: ND = Not Detectable.

## Data Availability

Data are contained within the article.

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
