# Peer review of "Pea Albumin Attenuates Dextran Sulfate Sodium-Induced Colitis by Regulating NF-κB Signaling and the Intestinal Microbiota in Mice"

_nutrients, 2022, doi:10.3390/nu14173611_

Round 1
Reviewer 1 Report
Inflammatory bowel disease (IBD) one of the important global problems. Several factors contribute to the development of IBD which genetic and environmental factors. Gut microbiota and dietary antigens are important environmental factors that influence the development of IBD. In this manuscript Zhang et al have shown that pea albumin has a alleviated the symptoms caused by DSS-induced colitis by suppressing inflammation and having a beneficial effect on gut microbes. Authors have done good work on characterizing composition of pea albumin. But the experimental methods were not described in detail. When it comes to mechanism of how pea albumin is alleviating DSS-induced inflammation authors tried to argue that pea albumin has a modulatory effect on both microbiota and inflammation. And none of them have good supporting data.
Minor comments:
1. Figure legend in the figure 1B, IC are not visible clearly.
2. Acronyms DSS+PL, PM, PH were used. Elaborate PL, PM, PH.
3. Figure 5C, D F G legends are not visible.
Major Comment:
1. Immunofluorescence (IF) images were not convincing. Isotype controls are very important to establish that the specificity of the signal. Please include isotype controls for CD4, CD4, CD11b, F4 80 antibodies that were used in figure 3. Also the signal seems to be non-specific and it’s everywhere. You might want to redo it with Isotype controls or do a flowcytometry experiment to look at immune cell populations in DSS and DSS+ pea albumin groups.
2. STAT3 is downstream of IL-22 receptor signaling. Th17 cells play a crucial role in the colitis disease outcomes. Authors might want to show that IL-22/IL-17 cytokine. Levels from colonic lysates in the experimental conditions tested.
3. In figure 5B Shannon index is affected in control+P group itself affected. Does it mean pea albumin treatment in control mice affects microbial community diversity?
Author Response
Reviewer #1:
Inflammatory bowel disease (IBD) one of the important global problems. Several factors contribute to the development of IBD which genetic and environmental factors. Gut microbiota and dietary antigens are important environmental factors that influence the development of IBD. In this manuscript Zhang et al have shown that pea albumin has alleviated the symptoms caused by DSS-induced colitis by suppressing inflammation and having a beneficial effect on gut microbes. Authors have done good work on characterizing composition of pea albumin. But the experimental methods were not described in detail. When it comes to mechanism of how pea albumin is alleviating DSS-induced inflammation authors tried to argue that pea albumin has a modulatory effect on both microbiota and inflammation. And none of them have good supporting data.
Reply: Thanks for reviewing our manuscript and the constructive comments. Your comments and suggestions are of great value for our study. More information was added in experimental methods, please see details in Methods section.
Pea albumin, a relatively new type of plant protein with a well-balanced AA profile, accounts for approximately 18–25% of total pea protein and is rich in bioactive compounds that may synergistically act to exert anti-inflammatory and gut microbiota modulation. However, the underlying mechanisms of pea albumin in protecting against colitis was unclear. This manuscript revealed the modulatory effect of pea albumin on both microbiota and inflammation.
Also, we corrected the spelling errors, grammatical errors as well as figure legends. Our point-by-point responses are shown below and heighted in revised version. We hope that all the questions raised during the review process have been well-addressed in this version.
Minor comments:
- Figure legend in the figure 2B, 2C are not visible clearly.
Reply: Thanks for pointing out this. We have revised the figure legends according to your comments. Please see Figure 2 for details.
- Acronyms DSS+PL, PM, PH were used. Elaborate PL, PM, PH.
Reply: We are sorry for the unclear statement. We elaborated the information for PL, PM and PH in main text (Methos section) and figure legends.
- Figure 5C, D F G legends are not visible.
Reply: Thanks for pointing out this. We have revised the manuscript according to your comments. Please see Figure 5 for details.
Major Comment:
- Immunofluorescence (IF) images were not convincing. Isotype controls are very important to establish that the specificity of the signal. Please include isotype controls for CD4, CD8, CD11b, F4/80 antibodies that were used in Figure 3. Also the signal seems to be non-specific and it’s everywhere. You might want to redo it with Isotype controls or do a flowcytometry experiment to look at immune cell populations in DSS and DSS + pea albumin groups.
Reply: Thank you for pointing out this omission. We provided isotype controls and new immunofluorescence staining for CD4, CD8, CD11b and F4/80 in revised version. Please see details in Figure 3C and D. At the end of this experiment, colon tissues were stored at -80℃, so it is difficult to do a flowcytometry experiment again. In addition, immunofluorescence staining for CD4, CD8, CD11b and F4/80 are often used to indicate the infiltration of inflammatory cells in previous published papers (Mol Nutr Food Res. 2021 Aug;65(15):e2001065; J Nutr. 2021 Nov 2;151(11):3391-3399; J Nutr. 2020 May 1;150(5):1116-1125.).
- STAT3 is downstream of IL-22 receptor signaling. Th17 cells play a crucial role in the colitis disease outcomes. Authors might want to show that IL-22/IL-17 cytokine. Levels from colonic lysates in the experimental conditions tested.
Reply: Thanks for your suggestion. We detected IL-22 and IL-17 level in the colonic lysates as suggested, please see details in Figure 4B. In addition, IL-6, a classical proinflammatory cytokine, has been reported to activate STAT3, whose activation is associated with the innate immune responses, and the production of mucins in the intestinal epithelium. We provided the IL-6 concentration in the previous version.
- In figure 5B Shannon index is affected in control + P group itself affected. Does it mean pea albumin treatment in control mice affects microbial community diversity?
Reply: Pea albumin treatment mice affects α-diversity with no significance, the P-value is 0.12 between control group and Control_P group, indeed. Table 1 show the P-value among groups.
Table 1 P-value among groups
|
Group 1 |
Group 2 |
P-value |
|
Control |
Control_P |
0.123 |
|
Control |
DSS |
0.109 |
|
Control |
DSS_PM |
0.875 |
|
Control_P |
DSS |
0.982 |
|
Control_P |
DSS_PM |
0.055 |
|
DSS |
DSS_PM |
0.031 |

Reviewer 2 Report
In this manuscript entitled “Pea Albumin Attenuates Dextran Sulfate Sodium-Induced Colitis by Regulating NF-κB Signaling and the Intestinal Microbiota in Mice”, Zhang et al. investigated the effect of pea albumin in DSS-induced colitis model. The authors showed that the treatment of pea albumin suppressed gut inflammation assessed by body weight, colon length, DAI, histology, proinflammatory cytokine, and NFkB signaling. They also demonstrated that pea albumin influences gut microbiota, particularly Lactobacillus. Therefore, they concluded that pea albumin treatment has protective effect against DSS-induced acute colitis. Although this is an interesting study, some critical experiments are missing. As commented below, additional experimental attempts are needed to strengthen their findings.
Major comments
#1: A previous study has already reported the effect of pea albumin in DSS-induced colitis (PMID: 25626675). What are the new findings of this study compared to previous studies? Also, the authors should mention previous study in section of introduction.
#2: To objectively assess the severity of colitis, histology score should be evaluated.
#3: Immunofluorescence staining of CD4 and CD8 T cells are not suitable to show the abundance of these cells. In fact, CD4 and CD8 T cells should present in DSS-untreated mice; however, these are not stained in DSS-untreated mice. Flow cytometry is appropriate to evaluate CD4 and CD8 T cells.
#4: Did the author evaluate pro-inflammatory cytokine in colon tissues? Pro-inflammatory cytokine in the colon is better to assess the degree of inflammation than cytosines in the serum.
#5: In figure 2, author said that 0.375g/kg of pea albumin was most effective to suppress gut inflammation compared to 0.1875 g/kg and 0.75 g/kg. However, author used 0.75g/kg pea albumin in later figure. Why did the author use 0.75 g/kg for assessment of histology, proinflammatory cytokine, and gut microbiota? Authors need to check whether the amount of pea albumin in figure legend is suitable.
Minor comments
#1: Higher resolution of figure is required.
#2: Line 206-208 should be deleted.
#3: In Table 1, histidine, tyrosine, and cysteine are not essential amino acid.
#4: Figure 2A and 2D-2F are missing in main text.
#5: Abbreviations of PL, PM, and PH are hard to understand. Showing dosing amount is more helpful for readers.
#6: The picture of Figure 2D is broken, so it’s hard to see the whole picture. A picture that shows the length of the colon is desirable.
Author Response
Reviewer #2:
In this manuscript entitled “Pea Albumin Attenuates Dextran Sulfate Sodium-Induced Colitis by Regulating NF-κB Signaling and the Intestinal Microbiota in Mice”, Zhang et al. investigated the effect of pea albumin in DSS-induced colitis model. The authors showed that the treatment of pea albumin suppressed gut inflammation assessed by body weight, colon length, DAI, histology, proinflammatory cytokine, and NF-κB signaling. They also demonstrated that pea albumin influences gut microbiota, particularly Lactobacillus. Therefore, they concluded that pea albumin treatment has protective effect against DSS-induced acute colitis. Although this is an interesting study, some critical experiments are missing. As commented below, additional experimental attempts are needed to strengthen their findings.
Reply: Thanks for reviewing our manuscript and the constructive comments. Your comments and suggestions are of great value for our study. Our point-by-point responses are shown below and heighted in revised version.
Major comments
#1: A previous study has already reported the effect of pea albumin in DSS-induced colitis (PMID: 25626675). What are the new findings of this study compared to previous studies? Also, the authors should mention previous study in section of introduction.
Reply: Thanks for your suggestion. In the present study, the results revealed the alleviatory mechanism of pea albumin in colitis through abrogation of overactivation of NF-κB and STAT3 signaling pathways as well as retortion of microbiota structure and composition with high-throughput sequencing when compared to previous study. Also, we introduced the previous study in the introduction and discuss section, please see details in line 69, 548, and 552.
#2: To objectively assess the severity of colitis, histology score should be evaluated.
Reply: Thanks for your suggestion. Histology scores were evaluated and provided in supplementary material (Figure 1) in this revised manuscript.
#3: Immunofluorescence staining of CD4 and CD8 T cells are not suitable to show the abundance of these cells. In fact, CD4 and CD8 T cells should present in DSS-untreated mice; however, these are not stained in DSS-untreated mice. Flow cytometry is appropriate to evaluate CD4 and CD8 T cells.
Reply: Thank you for pointing out this. We provided isotype controls and new immunofluorescence staining for CD4, CD8, CD11b and F4/80 in revised version. Please see details in Figure 3C and D. At the end of this experiment, colon tissues were stored at -80℃, so it is difficult to do a flowcytometry experiment again. In addition, immunofluorescence staining for CD4, CD8, CD11b and F4/80 are often used to indicate the infiltration of inflammatory cells in previous published papers (Mol Nutr Food Res. 2021 Aug;65(15):e2001065; J Nutr. 2021 Nov 2;151(11):3391-3399; J Nutr. 2020 May 1;150(5):1116-1125.).
#4: Did the author evaluate pro-inflammatory cytokine in colon tissues? Pro-inflammatory cytokine in the colon is better to assess the degree of inflammation than cytosines in the serum.
Reply: Thanks for this suggestion. We measured IL-17 and IL-22 concentration in colon tissue. We measured mRNA expression of proinflammatory cytokines (TNF-α, IFN-γ, IL-1β, IL-6, IL-17A, and IL-22) in the revised version. Please see Figure 4B, C for details.
#5: In figure 2, author said that 0.375g/kg of pea albumin was most effective to suppress gut inflammation compared to 0.1875 g/kg and 0.75 g/kg. However, author used 0.75g/kg pea albumin in later figure. Why did the author use 0.75 g/kg for assessment of histology, proinflammatory cytokine, and gut microbiota? Authors need to check whether the amount of pea albumin in figure legend is suitable.
Reply: Sorry for this big mistake. 0.75 g/kg of pea albumin was most effective to suppress gut inflammation compared to 0.375 g/kg and 1.5 g/kg. So, we used 0.75 g/kg pea albumin in later analysis.
Minor comments
#1: Higher resolution of figure is required.
Reply: Thanks for pointing out this. We provided higher resolution of figures in the revised version. Please see Figures for details.
#2: Line 206-208 should be deleted.
Reply: Thanks for your suggestion. We have deleted line 206-208 as suggested.
#3: In Table 1, histidine, tyrosine, and cysteine are not essential amino acid.
Reply: Sorry for the mistake, we have corrected it.
#4: Figure 2A and 2D-2F are missing in main text.
Reply: Thank you for pointing out this omission. We have revised the manuscript according to your comments. Please see details in line 113, 256, and 258.
#5: Abbreviations of PL, PM, and PH are hard to understand. Showing dosing amount is more helpful for readers.
Reply: Thanks for your suggestion. We illustrated abbreviations of PL (pea albumin low dosage), PM (pea albumin medium dosage), and PH (pea albumin high dosage) in the figure legends. We hope these changes are helpful for readers.
#6: The picture of Figure 2D is broken, so it’s hard to see the whole picture. A picture that shows the length of the colon is desirable.
Reply: Thanks for pointing out this. The suggested changes have been made. Please see details in Figure 2D.

Round 2
Reviewer 1 Report
Authors have done good work to answer reviewer questions and concerns. But there is still one concerning figure (Figure 3 C and 3D). I see CD4 or CD8 staining in the epithelial cell compartment rather than in lamina propria compartment. It might require a demarcated dashed line to distinguish the compartments and show the CD4, CD8, CD11b and F4 80 signal in the lamina propria compartment.
Apart from this everything looks good.
Author Response
Reviewer#1
- Authors have done good work to answer reviewer questions and concerns. But there is still one concerning figure (Figure 3 C and 3D). I see CD4 or CD8 staining in the epithelial cell compartment rather than in lamina propria compartment. It might require a demarcated dashed line to distinguish the compartments and show the CD4, CD8, CD11b and F4 80 signal in the lamina propria compartment.
Reply: Thanks for reviewing our manuscript and the constructive comments. We refreshed immunofluorescence staining images for CD4, CD8, CD11b and F4/80 and removed these results to supplementary material. Please see details in revised version.
- Apart from this everything looks good.
Reply: Thanks for your hard work.

Reviewer 2 Report
The authors revised the manuscript according to Reviewer’s comments. Although they revised manuscript well, but there are still concern about scientific issues as below.
#1: Authors assessed histological score as Reviewer suggested. Since this score is critical information for severity of colitis, authors should show the data in main figure.
#2: As Reviewer 1 also concerned, immunofluorescence staining of CD4, CD8, CD11b, and F4/80 are not standard method. Indeed, these cells should be present in control mice, but there are no positive cells in DSS-intreated group (Figure 3C). If authors still want to use immunofluorescence data, they should get other representative images or count positive cells for objective evaluation
#3: Figure 6 is broken.
Author Response
Reviewer#2
#1: Authors assessed histological score as Reviewer suggested. Since this score is critical information for severity of colitis, authors should show the data in main figure.
Reply:
We have moved histopathological scores to the main text (Figure 3B) as your suggested. Please see details in main text and Figure 3B.
#2: As Reviewer 1 also concerned, immunofluorescence staining of CD4, CD8, CD11b, and F4/80 are not standard method. Indeed, these cells should be present in control mice, but there are no positive cells in DSS-intreated group (Figure 3C). If authors still want to use immunofluorescence data, they should get other representative images or count positive cells for objective evaluation
Reply:
Thanks for reviewing our manuscript and the constructive comments. We refreshed immunofluorescence staining images of CD4, CD8, CD11b and F4/80 and removed these results to supplementary material (Figure S1). Please see details in revised version.
#3: Figure 6 is broken.
Reply:
Sorry for the mistake, we have corrected it.
